# Linking Neural Collapse and L2 Normalization with Improved Out-of-Distribution Detection in Deep Neural Networks

**Jarrod Haas**                                                                                           *jhaas@sfu.ca*
*SARlab, Department of Engineering Science*
*Simon Fraser University; Digitalist Group Canada*

**William Yolland**                                                                                       *yollandw@gmail.com*
*MetaOptima*

**Bernhard Rabus**                                                                                        *bernhard_t_rabus@sfu.ca*
*SARlab, Department of Engineering Science*
*Simon Fraser University*

**Reviewed on OpenReview:** *https://openreview.net/forum?id=fjkN5Ur2d6*

## Abstract

We propose a simple modification to standard ResNet architectures–L2 normalization over feature space–that substantially improves out-of-distribution (OoD) performance on the previously proposed Deep Deterministic Uncertainty (DDU) benchmark. We show that this change also induces early Neural Collapse (NC), an effect linked to better OoD performance. Our method achieves comparable or superior OoD detection scores and classification accuracy in a small fraction of the training time of the benchmark. Additionally, it substantially improves worst case OoD performance over multiple, randomly initialized models. Though we do not suggest that NC is the sole mechanism or a comprehensive explanation for OoD behaviour in deep neural networks (DNN), we believe NC's simple mathematical and geometric structure can provide a framework for analysis of this complex phenomenon in future work.

## 1 Introduction

It is well known that Deep Neural Networks (DNNs) lack robustness to distribution shift and may not reliably indicate failure when receiving out of distribution (OoD) inputs (Rabanser et al., 2018; Chen et al., 2020). Specifically, networks may give confident predictions in cases where inputs are completely irrelevant, e.g. an image of a plane input into a network trained to classify dogs or cats may produce high confidence scores for either dogs or cats. This inability for networks to "know what they do not know" hinders the application of machine learning in engineering and other safety critical domains (Henne et al., 2020).

A number of recent developments have attempted to address this problem, the most widely used being Monte Carlo Dropout (MCD) and ensembles (Gal and Ghahramani, 2016; Lakshminarayanan et al., 2017). While supported by a reasonable theoretical background, MCD lacks performance in some applications and requires multiple forward passes of the model after training (Haas and Rabus, 2021; Ovadia et al., 2019). Ensembles can provide better accuracy than MCD, as well as better OoD detection under larger distribution shifts, but require a substantial increase in compute (Ovadia et al., 2019).

These limitations have spurred interest in deterministic and single forward pass methods. Notable amongst these is Deep Deterministic Uncertainty (DDU) (Mukhoti et al., 2021). DDU is much simpler than many competing approaches (Liu et al., 2020; Van Amersfoort et al., 2020; van Amersfoort et al., 2021), produces competitive results and has been proposed as a benchmark for uncertainty methods. A limitation, as

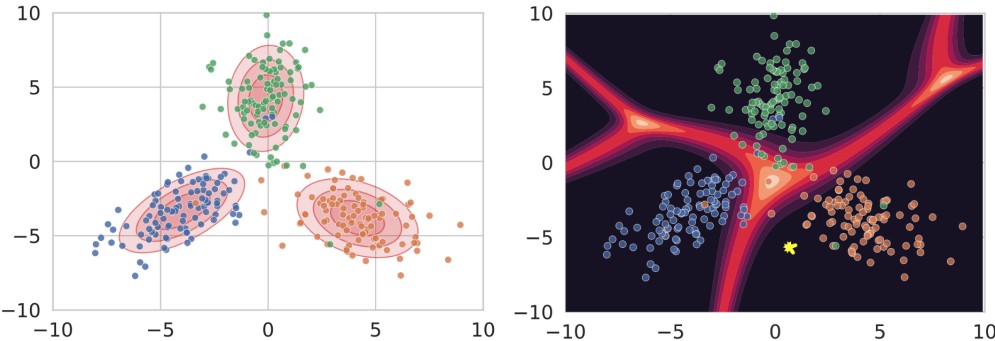

Figure 1: An illustration of the DDU method from Mukhoti et al. (2021) Left: In this hypothetical example with a two-dimensional feature space, DDU fits Gaussians over each of three classes as the components of a GMM, $q(y, z)$. Right: with standard decision boundaries (red), embeddings in this space that are far (yellow points) from class centroids are labelled with high confidence (darker areas are higher confidence).

shown in our experiments, is that DDU requires long training times and produces models with inconsistent performance.

We demonstrate that DDU can be substantially improved via L2 normalization over feature space in standard ResNet architectures. Beyond offering performance gains in accuracy and OoD detection, L2 normalization induces neural collapse (NC) much earlier than standard training. NC was recently found to occur in many NN architectures when they are overtrained (Papyan et al., 2020). This may provide a way to render the complexity of deep neural networks more tractable, such that they can be analyzed through the relative geometric and mathematical simplicity of simplex Equiangular Tight Frames (simplex ETF) (Mixon et al., 2022; Zhu et al., 2021; Lu and Steinerberger, 2020; Ji et al., 2021). Although this simplex ETF is limited to the feature layer and decision classifier, these layers summarize a substantial amount of network functionality. While Papyan et al. demonstrate increased adversarial robustness under NC, to the best of our knowledge, we present the first study of the relationship between OoD detection and NC.

We summarize our contributions as follows:

1) L2 normalization over the feature space of deep learning models results in OoD detection and classification performance that is competitive with or exceeds performance of the DDU benchmark. Most notably, the worst case OoD detection performance across model seeds is substantially improved.

2) Models trained with L2 normalization over feature space produce the aforementioned performance benefits in 17% (ResNet18) to 29% (ResNet50) of the training time of the DDU benchmark. Our proposed L2 normalization does not add any significant training time versus models without it.

3) L2 normalization over feature space induces NC as much as five times faster than standard training. Controlling the rate of NC may be useful for analyzing DNN behaviour.

4) NC is linked with OoD detection under our proposed modification to the DDU method. We show evidence that fast NC plays a role in achieving OoD detection performance with less training, and that training directly on NC has a substantially different effect on OoD performance than standard cross entropy (CE) training. This connection between simplex ETFs that naturally arise in DNNs and OoD performance permits an elegant analytical framework for further study of the underlying mechanisms that govern uncertainty and robustness in DNNs.

| | AUROC | | | | | | | | | Accuracy | | |
|---|---|---|---|---|---|---|---|---|---|---|---|---|
| | SVHN | | | CIFAR100 | | | Tiny ImageNet | | | CIFAR10 Test | | |
| **ResNet18** | No L2 350 | No L2 60 | L2 60 | No L2 350 | No L2 60 | L2 60 | No L2 350 | No L2 60 | L2 60 | No L2 350 | No L2 60 | L2 60 |
| Min | 0.877 | 0.848 | **0.924** | 0.861 | 0.796 | **0.878** | 0.881 | 0.809 | **0.898** | **0.928** | 0.881 | 0.924 |
| Max | 0.949 | 0.953 | **0.961** | 0.885 | 0.845 | **0.886** | 0.909 | 0.872 | **0.911** | **0.941** | 0.912 | 0.929 |
| Mean | 0.915±.018 | 0.912±.035 | **0.938**±.010 | 0.875±.008 | 0.824±.014 | **0.881**±.002 | 0.894±.009 | 0.841±.015 | **0.904**±.004 | **0.935**±.005 | 0.898±.008 | 0.927±.001 |
| **ResNet50** | No L2 350 | No L2 100 | L2 100 | No L2 350 | No L2 100 | L2 100 | No L2 350 | No L2 100 | L2 100 | No L2 350 | No L2 100 | L2 100 |
| Min | 0.903 | 0.869 | **0.927** | 0.817 | 0.794 | **0.892** | 0.881 | 0.852 | **0.912** | 0.930 | 0.896 | **0.937** |
| Max | **0.987** | 0.980 | 0.955 | 0.891 | 0.866 | **0.896** | 0.909 | 0.905 | **0.918** | **0.946** | 0.927 | 0.943 |
| Mean | **0.955**±.026 | 0.944±.029 | 0.945±.007 | 0.871±.018 | 0.837±.022 | **0.894**±.001 | 0.894±.020 | 0.880±.016 | **0.915**±.002 | 0.939±.006 | 0.916±.008 | **0.941**±.002 |

Table 1: OoD detection and classification accuracy results for ResNet18 and ResNet50 models, 15 seeds per experiment, trained on CIFAR10, and SVHN, CIFAR100 and Tiny ImageNet test sets used as OoD data. For all models, we indicate whether L2 normalization over feature space was used (L2/No L2) and how many training epochs occurred (60/100/350), and compare against the DDU baseline (No L2 350). Note that the variability of AUROC scores is substantially reduced under L2 normalization of feature space. With much less training, worst case OoD performance across model seeds improves substantially over the baseline, and mean performance improves or is competitive in all cases.

## 2 Background

### 2.1 Problem Definition

A standard classification model maps images to classes $f : x \to y$, where $x \in X$, the set of images, and $y \in \{1, ..., k\}$, the set of $k$ possible classes. The model is composed of a feature extractor which embeds images into feature space embeddings $z$ in $\mathbb{R}^d$ where $d$ is the size of the feature vector, and a linear decision classifier that transforms $z$ into a vector of length $k$, commonly known as the logits. These logits are run through a softmax function to produce a probability over classes over which the $argmax$ function identifies the class prediction $y$. We note that the set of images can be broken into $X \in X_{in}$ which are images drawn from the same distribution as the training data, and $X \in X_{out}$, which are all other images.

To evaluate models, we merge ID and OoD images into a single test set. OoD performance is then a binary classification task, where we measure how well OoD images can be separated from ID images using a score derived from our model. Our scoring rule is derived from a Gaussian Mixture Model (GMM) fit over $z$ (as discussed below) rather than the softmax outputs used for classification. We retain softmax outputs as we want to make sure that our ability to distinguish ID and OoD images does not hinder our ability to classify ID images.

### 2.2 Related Work

Until recently, most research toward uncertainty estimation in deep learning took a Bayesian approach. The high number of parameters in DL models renders posterior integration intractable, so approximations (typically variational inference) have been used (Gal and Ghahramani, 2016). Monte Carlo Dropout (MCD), which draws a connection between test time dropout and variational inference, has emerged as a popular method to estimate uncertainty (Gal and Ghahramani, 2016). Despite MCD's simplicity, scalability and applicability to nearly any model architecture, it is often outperformed by deterministic ensembles (Ovadia et al., 2019). Lakshminarayanan et al. observed that a simple average over the predictions of multiple deterministic models with the same architecture but starting from unique parameter initialisations produces a competitive uncertainty estimate (Lakshminarayanan et al., 2017).

The extra compute required for running multiple model passes at test time and for training and testing ensembles has motivated recent research around single model, single pass uncertainty estimates. Van Amersfoort et al. enforce a bi-Lipschitz penalty on gradients during training to create a distance-aware feature space in their Deep Uncertainty Quantification method (DUQ), which is then measured with a radial basis function (RBF) instead of a standard linear classifier (Van Amersfoort et al., 2020). While showing some promise, bi-lipschitz loss penalties can be unstable during training (Mukhoti et al., 2021). In later work, van Amersfoort et al. (2021) replace the RBF apparatus with a deep kernel and a point Gaussian Process (GP), while maintaining distance-awareness in feature space by adding spectral normalization throughout the model. A

similar approach by Liu et al. (2020) uses a Random Fourier Feature approximation to construct the GP (called Spectral-normalized Neural Gaussian Process, or SNGP), but results are not as competitive. Lee et al. (2018) propose fitting a class-wise conditional Gaussian with shared covariance matrix to multiple layers, but employ OoD and adversarial data to learn a weighted average of scores over layers. They also add noise to inputs at test time to enhance OoD separability. ODIN produces competitive results, but also requires OoD data for hyperparameter tuning (Liang et al., 2017). Generalized ODIN relaxes this requirement (Hsu et al., 2020), but both versions require an additional backward pass at test time to perturb inputs. Finally, Hendrycks and Gimpel (2016) proposed simply measuring maximum softmax scores of converged models as a benchmark for OoD detection. We believe that OoD detection has moved well past what out-of-the-box, uncalibrated softmax scores can accomplish, and that Deep Deterministic Uncertainty provides a better benchmark for the evaluation of single pass methods (Mukhoti et al., 2021).

## 2.3 Deep Deterministic Uncertainty

Deep Deterministic Uncertainty (DDU) was proposed by Mukhoti et al. as a simple but very competitive benchmark to detect OoD samples (Mukhoti et al., 2021). It requires only a single pass through a single network, and empirically demonstrates that more complex methods such as DUQ and SNGP are not necessary. To evaluate uncertainty, it uses a class-wise Gaussian Mixture Model (GMM) $q(y, z)$, with mean and covariance parameters retrieved from the feature layer via a one-time pass over the training set using a converged model. At test time, extracted features $z$ are evaluated under the GMM as $q(z) = \sum_y q(z|y)q(y)$. This log probability is used as the scoring method to assess separability of ID and OoD data.

Mukhoti et al.'s approach is motivated by a simple idea: feature space must be "well regularized", i.e. images that are very different in input space are different in feature space ("sensitivity"), and images that are similar in input space are similar in feature space ("smoothness"). This can be interpreted as enforcing a bi-Lipschitz constraint over feature space:

$$K_L d_I(x_1, x_2) \leq d_F(f\theta(x_1), f\theta(x_2)) \leq K_U d_I(x_1, x_2) \tag{1}$$

for feature extractor $f$ with parameters $\theta$, inputs $x_1, x_2$, distance metrics $d_I, d_f$ over input space and feature space respectively, and lower and upper bounds $K_L$ (sensitivity) and $K_U$ (smoothness), respectively (Mukhoti et al., 2021; Liu et al., 2020). Leaky ReLUs and strided average pooling of 1x1 convolutions (to perform downsampling in residual connections) can improve sensitivity, while spectral normalization can improve smoothness and encourage bi-Lipschitz continuity in residual networks (Mukhoti et al., 2021; Smith et al., 2021).

Under a well-regularized feature space arising from a converged model, OoD images should be mapped less often to ID feature space. OoD detection is then a matter of measuring the distance of an extracted feature $f\theta(x)$ from dense, class-wise feature-space clusters that emerge during training. Additionally, fitting a GMM over these clusters prevents the network from assigning high confidence to points far away from the class clusters simply because they fall within a decision region (Figure 1).

Importantly, if a feature extractor is too sensitive, ID feature vectors may not cluster tightly enough, and ID images would then get flagged as OoD due to their distance from modes in the GMM. If too smooth, the feature extractor won't pick up on differences between OoD and ID images, and may place OoD images too close to GMM modes.

Our ablation study results (Table 2) show that L2 normalization over feature space improves OoD detection when used in addition to the bi-Lipshitz-enforcing regularization techniques proposed in the DDU benchmark. In Section 4, we show an empirical link between L2 normalization, Neural Collapse and OoD detection performance.

## 2.4 L2 Normalization of Feature Space and Neural Collapse

DDU works well under the assumption of a well-regularized feature space. However, the bi-Lipschitz continuity encouraged by the aforementioned methods affords no explicit structure over feature space. We suggest

| Intervention | | | AUROC | | | Accuracy | | |
|---|---|---|---|---|---|---|---|---|
| Spectral Norm | Leaky/GAP | L2-Norm | Min | Max | Mean | Min | Max | Mean |
| ✗ | ✗ | ✗ | 0.855 | 0.939 | 0.910 | 0.884 | 0.927 | 0.916 |
| | | ✓ | 0.894 | 0.951 | 0.932 | 0.924 | 0.930 | 0.927 |
| | ✓ | ✗ | 0.830 | **0.965** | 0.889 | 0.872 | 0.909 | 0.898 |
| | | ✓ | 0.917 | 0.958 | **0.944** | **0.925** | 0.929 | 0.927 |
| ✓ | ✗ | ✗ | 0.884 | 0.954 | 0.923 | 0.904 | 0.926 | 0.917 |
| | | ✓ | 0.903 | 0.959 | 0.934 | **0.925** | **0.931** | **0.928** |
| | ✓ | ✗ | 0.848 | 0.953 | 0.912 | 0.881 | 0.912 | 0.898 |
| | | ✓ | **0.924** | 0.961 | 0.938 | 0.924 | 0.929 | 0.927 |

Table 2: ResNet18 ablation study of effects from the DDU benchmark along with our method, 15 seeds per experiment. Models trained on CIFAR10, with SVHN as OoD data. When combined with any other interventions, our method improves worst case and average AUROC scores. In many cases, minimum AUROC across all seeds is improved by several points.

that the simplex ETF structure arising from Neural Collapse improves sensitivity and smoothness and is thereby linked with better OoD detection results.

Although feature space is structured once model accuracy converges–this being a simple consequence of the decision layer's need to divide feature space in order to produce class predictions–the structure that emerges is limited and highly variable, even under bi-Lipschitz regularization techniques: class clusters have high non-uniform variance, and clusters are only separated to the degree that the decision layer can satisfy training demands for low cross-entropy loss (Figure 8). As a consequence, it is well known that data embedded far from class clusters can have high confidence scores (Hein et al. (2018), Figure 1). This is also evidenced by models with high accuracy and high-variance class clusters (Table 2, 3).

Given that we want to organize feature space in the manner most conducive to OoD detection, we suggest that following the principles of sensitivity and smoothness should be the aim: tightening clusters and placing them away from each other. Features in this proposed configuration would have low within-class variance (smoothness), and higher between-class variance (sensitivity). Intuitively, networks trained to adhere to this structure should excel at mapping known inputs to within these tight clusters, while unknown inputs should become difficult to map precisely and would tend to fall outside of the clusters and into the empty feature space around them.

Although there are many possible ways to structure feature space, a reasonable starting point is a hypersphere. Numerous works exist that seek to exploit the properties of hyperspheres in order to make high-dimensional information-comparison problems, like OoD detection, more tractable (Sablayrolles et al., 2018; Zhou et al., 2020; Zheng et al., 2022; Liu et al., 2017). An interesting property of a hypersphere on $R^D$ is that if $D$ is sufficiently larger than the number of points $N$ that one wishes to embed on its surface, the maximum distance between all points is obtained when all point-wise distances are equal. In other words, constraining feature space to a hypersphere allows a structure over which class clusters could be maximally and equally distanced from one another, providing substantial unused space between clusters. We note that fully maximizing between-class distance is probably not ideal, as this would rigidly enforce too much sensitivity i.e. we would like the feature extractor to have some flexibility in placing more similar classes closer together. A balance must be struck between spreading classes out and respecting appropriate Lipschitz bounds.

Papyan et al. (2020) recently observed a phenomenon known as Neural Collapse (NC) which progresses as networks are trained, ultimately leading to feature space and decision space conforming to a structure known as a simplex equiangular tight-frame (see Section 3.2 for definition and measurement). One of the key properties of NC is variability collapse, wherein class clusters in embedding space collapse toward a single point in feature space. Additionally, as feature space convergences to a simplex ETF, clusters are positioned

| Intervention | | | NC1 | | NC2 EA Means | | NC2 EA Class | | NC2 EN Means | | NC2 EN Class | | NC3 | | NC4 | |
|---|---|---|---|---|---|---|---|---|---|---|---|---|---|---|---|---|
| Spectral Norm | Leaky ReLU | L2-Norm | Mean | SD | Mean | SD | Mean | SD | Mean | SD | Mean | SD | Mean | SD | Mean | SD |
| No SN | No Leaky | No L2 | 1.967 | 0.136 | 0.076 | 0.007 | 0.033 | 0.011 | 0.098 | 0.008 | 0.094 | 0.005 | 0.169 | 0.015 | 0.002 | 0.001 |
| | | L2 | 0.154 | 0.004 | 0.014 | 0.001 | 0.023 | 0.001 | 0.044 | 0.002 | 0.083 | 0.001 | 0.038 | 0.001 | 0.000 | 0.000 |
| | Leaky | No L2 | 2.558 | 0.329 | 0.100 | 0.018 | 0.064 | 0.027 | 0.118 | 0.017 | 0.112 | 0.030 | 0.271 | 0.059 | 0.008 | 0.005 |
| | | L2 | 0.170 | 0.005 | 0.013 | 0.001 | 0.024 | 0.001 | 0.047 | 0.002 | 0.084 | 0.001 | 0.039 | 0.001 | 0.000 | 0.000 |
| SN | No Leaky | No L2 | 2.139 | 0.412 | 0.083 | 0.019 | 0.036 | 0.013 | 0.108 | 0.018 | 0.095 | 0.006 | 0.182 | 0.029 | 0.004 | 0.004 |
| | | L2 | 0.163 | 0.005 | 0.015 | 0.002 | 0.024 | 0.001 | 0.047 | 0.002 | 0.085 | 0.001 | 0.038 | 0.001 | 0.000 | 0.000 |
| | Leaky | No L2 | 2.690 | 0.368 | 0.111 | 0.018 | 0.051 | 0.020 | 0.124 | 0.015 | 0.106 | 0.012 | 0.260 | 0.031 | 0.008 | 0.005 |
| | | L2 | 0.171 | 0.005 | 0.014 | 0.002 | 0.023 | 0.001 | 0.049 | 0.003 | 0.084 | 0.001 | 0.039 | 0.001 | 0.000 | 0.000 |
| SN, Leaky, No L2 - 350 Epochs | | | 0.393 | 0.164 | 0.024 | 0.003 | 0.025 | 0.006 | 0.041 | 0.005 | 0.025 | 0.002 | 0.044 | 0.004 | 0.000 | 0.000 |

Table 3: Neural Collapse measurements for ResNet18 ablation study and ResNet18 DDU baseline. 15 seeds for all experiments, ablation models trained for 60 epochs, DDU baseline trained for 350 epochs. Lower numbers indicate more NC. Most metrics are an order of magnitude lower with the L2 intervention. Models trained for 350 epochs slightly less but similar amounts of NC as L2 models trained for only 60 epochs.

at maximally equiangular distances from each other. In other words, DNNs gradually and automatically proceed toward the hypersphere structure we seek (Figure 2).

The difficulty here is that NC fully converges only after the terminal phase of training, i.e. after cross-entropy loss goes to zero. This means that networks need to be substantially over-trained before feature space becomes structured in a manner which we can fully exploit.

Since the Simplex ETF structure also results in class clusters existing on the surface of a hypersphere (a consequence of equinormality, see Section 3.2), we hypothesize that it is possible to induce NC more quickly by simply constraining feature space to a hypersphere from the onset of training. L2 normalization over feature space constrains feature vectors to a point on the surface of a hypersphere by restricting the magnitude of all feature vectors to be uniform:

$$z_{norm} = \frac{z}{max(\| z \|_2, \epsilon)} \tag{2}$$

where $R^d$ is the space of real numbers over $d$ dimensions and $\epsilon$ is added in the event of the zero norm. We note that L2 normalization of the feature space immediately constrains features to satisfy the equinormality property of a Simplex ETF (see the second property of NC in Section 3.2) from the onset of training. Since feature vector magnitude is no longer a discriminatory factor for the classifier, feature vectors must be differentiated entirely by angular distance. For this reason, we view L2 normalization as a regularizer, limiting the set of possible feature spaces to only those defined on the surface of a hypersphere. This expedites the progression of equiangularity as well as the other aspects of NC (Table 3). Importantly, this can be easily implemented in one line of code and does not require a a sophisticated loss function, parameter tuning, nor any monitoring of NC measurements. In this way, L2 normalization acts as a regularizer that constrains feature space to a hypersphere and expedites NC, and NC is linked with OoD detection under the DDU method by explicitly and progressively structuring feature space as a simplex ETF.

## 3 Methodology

### 3.1 Models and Loss Functions

For all experiments we used either ResNet18 or ResNet50 models provided with the DDU benchmark code (Mukhoti et al., 2021). All were trained from fifteen independent seeds (training details can be found in Appendix A.1). All baselines used a standard cross entropy (CE) objective function during training. The NC intervention group described in Section 4.2 did not use a use a CE loss, but instead used a loss function containing the differentiable metrics described below in Section 3.2:

$$L_{NC}(f(x)) = NC_1 + EN_{means} + EN_{classifier} + EA_{means} + EA_{classifier} + NC_3 \tag{3}$$

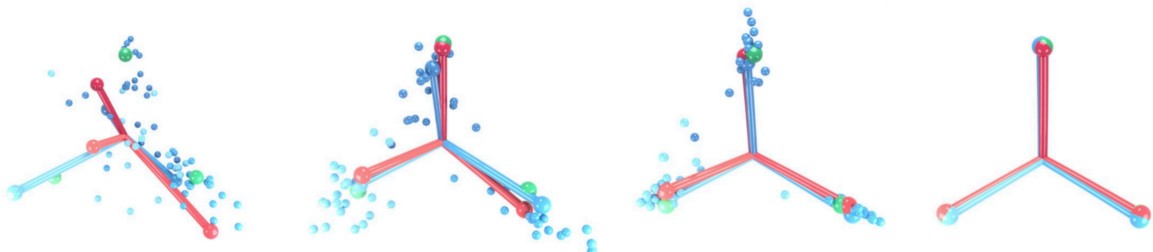

Figure 2: Progression of Neural Collapse during training (left to right). Small blue spheres represent extracted features (classes are different shades of blue), blue ball-and-sticks are class-means, red ball-and-sticks are linear classifiers. Features collapse to low-variance class means and linear classifiers align with these. Note that the simplex ETF pictured is on the 2D plane in 3D space, such that each arm is equidistant at 120 degrees. Image from (Papyan et al., 2020).

Note that the metric for $NC4$ is not used, as it requires an *argmin* function which is not differentiable. Although these metrics do not all have the same scale, they all proceed to zero, and we did not find it necessary to use any weighting scheme within the loss function.

### 3.2 Measuring Neural Collapse

NC has four properties:

NC1: Variability collapse: the within-class covariance of each class in feature space approaches zero.

NC2: Convergence to a Simplex Equiangular Tight Frame (Simplex ETF): the angles between each pair of class means are maximized and equal and the distances of each class mean from the global mean of classes are equal, i.e. class means are placed at maximally equiangular locations on a hypersphere.

NC3: Convergence to self-duality: model decision regions and class means converge to a symmetry where each class mean occupies the center of it's decision region, and all decision regions are equally sized.

NC4: Simplification to Nearest Class Center (NCC): the classifier assigns the highest probability for a given point in feature space to the nearest class mean.

Papyan et al. (2020) use seven different metrics (Eq. 3 to Eq. 8) to observe these properties. All are differentiable and used in the NC loss function for the experiment in Section 4, except for the $NC4$ metric (Eq. 9), which is not differentiable.

The within-class variance, $NC1$, is measured by comparing the within-class variance to the between-class variance,

$$NC1 = Tr\{\Sigma_W \Sigma_B^\dagger \backslash C\} \tag{4}$$

where $Tr$ is the trace operator, $[\cdot]^\dagger$ indicates the Moore-Penrose pseudoinverse, and $C$ is the number of classes. Although Papyan et al. (2020) could simply have used the trace of $\Sigma_W$ to measure variability collapse, we speculate they incorporated $\Sigma_B^\dagger$ (i.e. the between-class precision) because it becomes smaller as the class-means separate from each other around the hypersphere. Note that $NC1$ is not necessarily zero if cross-entropy loss is zero: as shown in Figure 1, a class can have multiple areas where class confidence is maximized.

$NC2$ is indicated through four measurements. The equinormality of class means and classifier means is given by their coefficient of variation,

$$EN_{means} = \frac{std_c(\parallel u_c - u_G \parallel_2)}{avg_c(\parallel u_c - u_G \parallel_2)} \tag{5}$$

$$EN_{classifier} = \frac{std_c(\parallel w^T \parallel_2)}{avg_c(\parallel w^T \parallel_2)} \tag{6}$$

where $u_c, u_G$ in $\mathbb{R}^d$ are the class means and global mean of classes, $std$ and $avg$ are standard deviation and average operators, and $\parallel \cdot \parallel_2$ is the L2-norm operator. The global class mean is computed as the mean of all class clusters, and class means themselves are each the mean of all feature vectors generated by the model for images of that particular class.

Maximum equiangularity is measured via the Gram matrix of normalized class means (or classifier means), $u_c$. The Gram matrix $G = \parallel u_c \parallel_F^T \cdot \parallel u_c \parallel_F$ of class means (where $\parallel \parallel_F$ is the Frobenius norm, $u_c$ contains columnwise class means) contains, in its off-diagonal elements, the angles between each pair of class mean vectors. Maximum cosine distance between $C$ classes is $-1/(C-1)$ (assuming that the dimension of the embedding space is high enough to embed all vectors at equal angles from each other). We thus add this quantity to each element of the mutual coherence matrix, since all elements will then go to zero as we converge to equiangularity. Finally, we assign zero to the diagonal, as we do not need the distance of vectors with themselves, and we have

$$EA_{means} = \frac{\parallel G + C_{cos} - diag\{G + C_{max}\} \parallel_1}{C * (C-1)} \tag{7}$$

where $G$ is the Gram matrix of normalized class means, $C_{cos}$ is the matrix whose elements are set to $-1/(C-1)$, and $diag$ retrieves the diagonal of a matrix. We take the L1-norm to map this quantity to a single number, and then normalize by the number of pairwise combinations.

The self-duality of class means and classifiers, $NC3$, is measured as the square of L2 normalized differences between classifier and class means,

$$NC_3 = \parallel \tilde{W}^T - \tilde{M} \parallel_2^2 \tag{8}$$

where $W$ is the matrix of last-layer classifier weights, $M$ is the matrix of class means, $\tilde{[\cdot]}$ denotes a matrix with unit normalized means over columns, and $\parallel \cdot \parallel_2^2$ denotes the square of the L2-norm.

Finally, Nearest Class Center classification, $NC4$, is measured as the proportion of training set samples that are misclassified when a simple decision rule based on the distance of the feature space activations to the nearest class is used,

$$NC_4 = argmin_c \parallel z - u_c \parallel_2 \tag{9}$$

where $z$ is the feature space vector for a given input.

## 4 Experiments

### 4.1 Faster and More Robust OoD Results

Our results in Table 1 demonstrate that L2 normalization over feature space produces results that are competitive with or exceed those obtained using the DDU benchmark, and are obtained in less training time for ResNet18 and ResNet50 models. For ResNet18, our mean Area Under the Receiver-Operator Curve (AUROC) scores exceed those of the baseline in only 60 epochs (17% of DDU baseline training time), with only a .008 reduction in classification accuracy. For ResNet50, we achieve higher mean accuracy than the

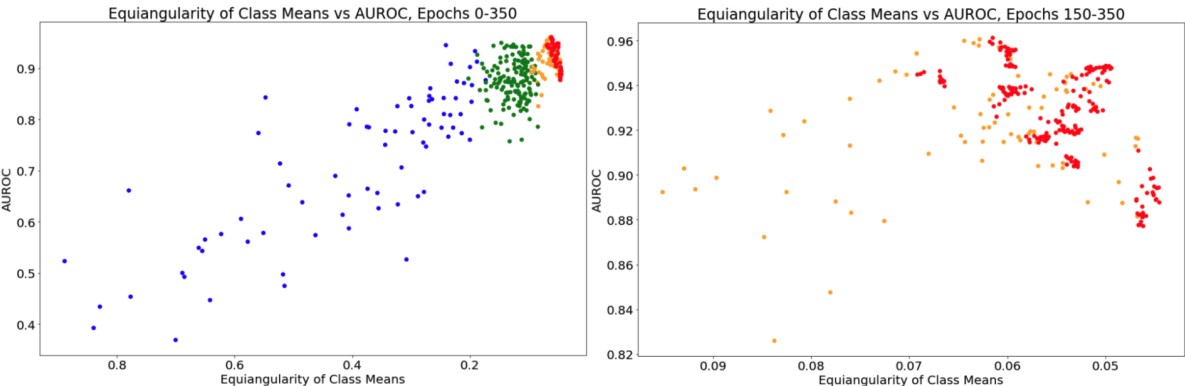

Figure 3: Equiangularity of class means ($NC2$) vs. AUROC scores during training for 15 ResNet18 seeds trained for 350 epochs (no L2 normalization). Each point is the score of an individual model at a particular epoch, every tenth epoch is captured. Colors are segmented by training epochs: blue 0-40, green 50-140, orange 150-190, and red 190-350. The Pearson R coefficient is -0.894 (NC is decreasing as AUROC increases). Right: only epochs 150-350 are shown (individual models tend to appear as clusters). Too much NC can have a slightly detrimental effect (less than 0.05 points AUROC on average) as we would expect, but there is very little risk in overtraining models, even without L2 normalization.

the baseline in only 100 epochs (29% of DDU baseline training time), while mean AUROC is lower by only 1 point.

Most notably, the lower bounds of OoD detection performance are significantly improved on both models (Table 1). For Resnet18, we improve the AUROC baseline by .047, and on ResNet50 we improve by .024. These effects are even more pronounced when compared with the same number of training epochs in the absence of L2 normalization. Note that while AUROC and classification scores might improve with more training, our scope here is to emphasize that a substantial reduction in compute time is possible with this technique. We achieve substantially better worst case OoD performance with only a fraction of the training time stated in the DDU benchmark. Notably, the variability of AUROC performance across models trained from different initializations is also heavily decreased with L2.

An ablation study shows that mean accuracy, minimum accuracy, and AUROC are all improved across each of the pairwise L2/no-L2 comparisons (Table 2). Furthermore, the worst performing models (min and mean AUROC) using L2 normalization still exceed the best performing models trained without L2 normalization across the same metrics.

We also assess how standard softmax scores perform when used as OoD decision scores. These results are detailed in Table 4 in Appendix A.2. Softmax scores perform worse in all cases, with the exception of the ResNet50, No L2 350 epoch model.

Finally, we note that a large amount of NC results in a feature space that contains most information along $k = numclasses$ dimensions, a small subset of the total size of the feature space (e.g. $\mathbb{R}^{512}$ in ResNet18). This raises the question of whether a GMM could simply be fit over logit space after substantial NC, since logit space is in $\mathbb{R}^k$. We provide and discuss experimental results showing that GMMs are best fit to feature space in Appendix A.3

## 4.2 Linking Neural Collapse with OoD Detection

Our intuition that L2 normalization over feature space would induce NC faster is empirically proven throughout our experiments. All aspects of NC are induced more rapidly, with nearly all measures showing greater NC in 60 epochs than the equivalent model counterparts trained for 350 epochs without L2 normalization (Table 3). We also note that in models without L2 normalization, NC is generally an order of magnitude less after 60 epochs than those with L2 normalization (Table 3).

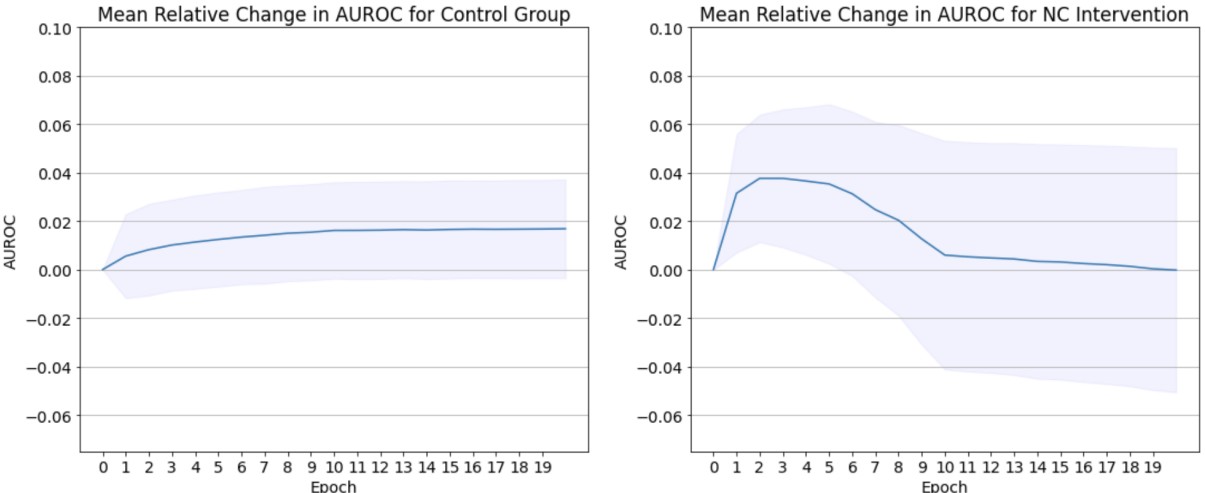

Figure 4: NC loss intervention experiment. Mean relative change in AUROC when training from the 50th epoch of a no L2 ResNet18 over 15 unique model seeds for an additional 20 epochs. Blue line is the mean over model seeds, blue shading is standard deviation. Left: The control group improves with more training as expected, but the difference is small. Right: The intervention group mean AUROC score improves at a peak of 0.38 after two epochs, 4.75x more than the control group mean improvement of .008 over the same training period. The onset of NC has a clear and immediate impact on AUROC outcomes.

Figure 3 shows the effect of training non-L2 normalized ResNet18s for 350 epochs on NC and AUROC scores. As expected, NC is strongly correlated with OoD performance. We also note evidence that too much NC can slightly reduce performance from peak (Figure 3, Right).

### 4.2.1 Neural Collapse Training Intervention

To further investigate the connection between NC and OoD performance, we use 15 random seeds of ResNet18 models trained for 350 epochs with no L2 normalization. We observe during training that after 50 epochs, accuracy and AUROC scores both approach noisy plateaus. We use this observation to create two groups, a control and an intervention. All models within these groups are initialized using the $50^{th}$ epoch of their respective seed. Both groups then begin training at an order of magnitude lower loss to encourage convergence, and both groups step down the loss after 10 epochs to control for the possible effect of learning rate. The control group models continue training using the standard cross entropy loss. The intervention group is trained using the differentiable NC metrics listed in Equation 3.

As shown in Figure 7 (Appendix A.5), models in the intervention group have substantially greater amounts of NC, as expected. CE loss and classification accuracy remain relatively stable in the intervention models, owing to the fact that CE is excluded from the loss function. The intervention group is thus disentangled from confounding influence from the learning rate or the CE objective. However, in Figure 4, we see that AUROC is substantially different between the intervention and control groups.

Models perform better under NC and these improved scores arrive with less training: after only 2 epochs. Intervention models improve AUROC an average of 0.038 points from intervention onset, 4.75x the average improvement of .008 from control models during this time. Even if allowed to train for the full 20 epochs, control models improve to a maximum average of .017.

### 4.2.2 Investigating Collapse

Figure 4 shows that too much training directly with NC loss hinders performance. This aligns with our intuition that too much NC can adversely effect the bi-Lipschitz constraint: extracted OoD features are not sufficiently differentiated from ID features and get collapsed into class clusters.

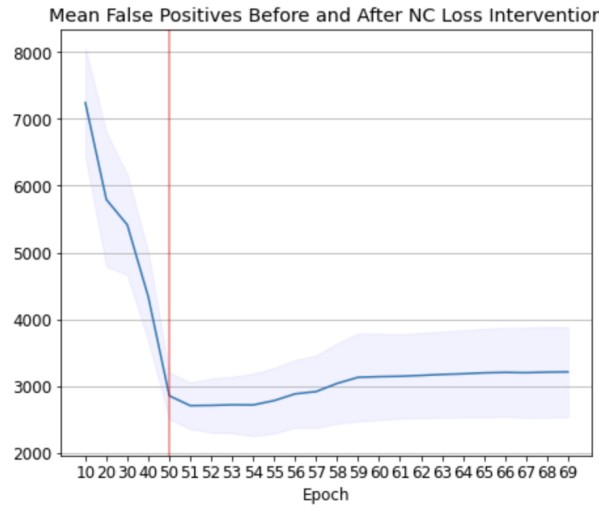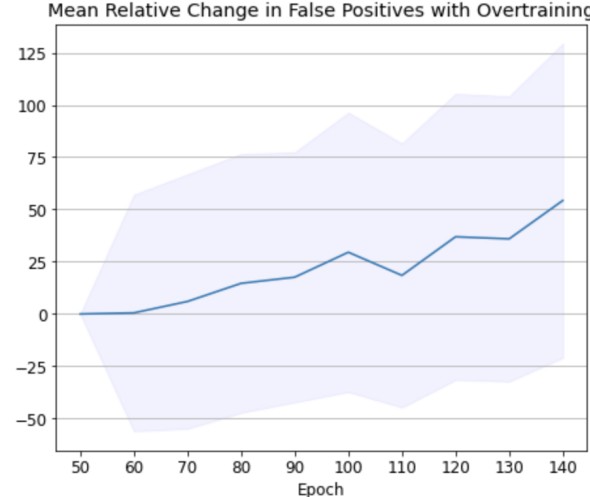

Figure 5: Neural Collapse induced by NC loss leads to false positive images collapsing into class means quickly, but this effect is heavily mitigated when NC is induced under CE loss. We know OoD images are collapsed into class means because the variability of these means is decreasing as training continues and as false positives increase. The blue line is the mean over 15 model seeds, blue shading is standard deviation. Left: Training on NC loss directly first decreases false positives (see also Figure 4), but too much NC starts to collapse OoD and ID features together. Within only 20 epochs, this results in over 500 more false positives. Right: A similar but far less pronounced trend is found if we overtrain our ResNet18 L2 normalized model from 60 to 150 epochs. In this case, false positives only increase by about 50: an order of magnitude less OoD collapse over five times as much training. L2 normalization thus allows models to be trained to convergence via standard loss and accuracy metrics without the need to monitor or tune NC.

To examine this, we measure the false positive detection rate before and during the NC intervention (Figure 5). As there are ten thousand ID test images with CIFAR10, false positives are defined as OoD images that have confidence scores in the top ten thousand of all scores. As expected, the false positive rate drops steadily up to and shortly beyond intervention onset, but then begins to increase again. Since we know that variability is collapsing (Figure 7) as the false positive rate increases, we know that OoD images are getting higher scores due to their collapse into class means along with ID images.

To see if a similar effect occurs under NC from CE loss, we over-train our L2 normalized ResNet18 model from 60 epochs through to 150 epochs. Figure 5 shows that the false positive rate remains stable for approximately 10 epochs before increasing nearly monotonically for the remainder of training. However, in this case the effect is an order of magnitude smaller than when we use NC loss. Instead of an extra 500 (on average) false positives within 20 epochs, the L2 case only increases by an average of 54 false positives after an extra 90 epochs of training. When over-training our no L2 baseline from 350 to 500 epochs, false positives increase by a maximum average of 20.

Using L2 normalization on feature space is quite stable, and peak or near-peak results can be obtained without much fine-tuning of training length. However, we do note that there is a trade off between classification accuracy and OoD detection performance. When substantially overtraining to 350 epochs with L2 normalization, accuracy scores improve slightly over the no L2 cases, but OoD performance takes a significant penalty (see Table 6, Appendix A.4). We thus note that our method is best used with short training schedules. Highest accuracy on validation can still be used to guide training, but shouldn't be used to determine optimal OoD performance.

Over-training directly with NC loss can have a large detrimental effect, despite it resulting in similar NC measurements to the counterpart CE loss cases. It is not optimal to simply induce NC on a pre-converged model via NC loss, although doing so can significantly boost AUROC scores. This suggests that CE loss, as well it's potential interactions with spectral norm and leaky ReLUs plays an important role in conditioning

feature space with an optimal bi-Lipschitz constraint. The benefit of this, however, is that our method is stable with respect to training–there is no need to monitor NC measurements in order to arrive at or near the peak of OoD performance or reap the benefits of much improved worst-cast performance.

Finally, we observe that while NC is directly correlated with improved OoD performance as explained above, per model amounts of NC are not directly correlated with peak AUROC. In other words, although optimal OoD detection happens within a similar range of NC measurements, specific amounts of NC cannot be used to precisely predict peak AUROC scores for a given model. Again, this harmonizes with our intuition that NC is only one of several factors acting to condition feature space for OoD detection. These factors are all susceptible to the variability of training dynamics, as well as situational dependencies such as architectures and datasets.

## 5  Conclusion and Future Work

We propose a simple, one-line-of-code modification of the Deep Deterministic Uncertainty benchmark that provides superior OoD detection and classification accuracy results in a fraction of the training time. We also establish that L2 normalization induces NC faster than regular training, and that NC is linked to OoD detection performance under the DDU method. Although we do not suggest that NC is the sole explanation for OoD performance, we do expect that its simple structure can provide insight into the complex and poorly understood behaviour of uncertainty in deep neural networks. We believe that this connection is a compelling area of future research into uncertainty and robustness in DNNs.

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

# A   Appendix

## A.1   Training Details

All models (except those explicitly noted in the ablation study) use spectral normalization, leaky ReLUs and Global Average Pooling (GAP), as these produce the strongest baselines. Each experiment was conducted with fifteen randomly initialized model parameter sets; no fixed seeds were used at any time for initialization. We set the batch size to 1024 for all training runs, except the NC intervention models, which were more stable when training with a batch size of 2048. All training was conducted on four NVIDIA V100 GPUs in PyTorch 1.10.1 Paszke et al. (2019).

Stochastic gradient descent (SGD) with an initial learning rate of $1e^{-1}$ was used as the optimizer for all experiments. We used a learning rate schedule that decreased by one order of magnitude at 150 and 250 epochs for the 350 epoch models, as per the DDU benchmark. We adjust the learning rate at 75 and 90 for the 100 epoch ResNet50 models, and at 40 and 50 for the 60 epoch ResNet18 models. Models were trained on the standard CIFAR-10 training data set with a validation size of 10% created with a fixed random seed.

## A.2   Effect of L2 Normalization on Softmax Scores for OoD Detection

| AUROC (GMM Over Feature Space) | | | | | | | | | | | |
|---|---|---|---|---|---|---|---|---|---|---|---|
| **SVHN** | | | | **CIFAR100** | | | | **Tiny ImageNet** | | | |
| **ResNet18** No L2 350 | L2 350 | No L2 60 | L2 60 | No L2 350 | L2 350 | No L2 60 | L2 60 | No L2 350 | L2 350 | No L2 60 | L2 60 |
| **Min** 0.877 | 0.879 | 0.848 | **0.924** | 0.861 | 0.870 | 0.796 | **0.878** | 0.881 | 0.885 | 0.809 | **0.898** |
| **Max** 0.949 | 0.921 | 0.953 | **0.961** | 0.885 | 0.879 | 0.845 | **0.886** | 0.909 | 0.902 | 0.872 | **0.911** |
| **Mean** 0.915±0.018 | 0.904±0.011 | 0.912±0.035 | **0.938**±0.010 | 0.875±0.008 | 0.874±0.002 | 0.824±0.014 | **0.881**±0.002 | 0.894±0.009 | 0.892±0.004 | 0.841±0.015 | **0.904**±0.004 |

| **ResNet50** No L2 350 | L2 350 | No L2 100 | L2 100 | No L2 350 | L2 350 | No L2 100 | L2 100 | No L2 350 | L2 350 | No L2 100 | L2 100 |
|---|---|---|---|---|---|---|---|---|---|---|---|
| **Min** 0.903 | 0.903 | 0.869 | **0.927** | 0.817 | 0.876 | 0.794 | **0.892** | 0.881 | 0.880 | 0.852 | **0.912** |
| **Max** **0.987** | 0.943 | 0.98 | 0.955 | 0.891 | 0.891 | 0.866 | **0.896** | 0.909 | 0.898 | 0.905 | **0.918** |
| **Mean** **0.955**±0.026 | 0.926±0.013 | 0.944±0.029 | 0.945±0.007 | 0.871±0.018 | 0.886±0.005 | 0.837±0.022 | **0.894**±0.001 | 0.894±0.020 | 0.888±0.005 | 0.880±0.016 | **0.915**±0.002 |

(a) The variability of AUROC scores is substantially reduced under L2 normalization of feature space. With much less training, worst case OoD performance across model seeds improves substantially over the baseline, and mean performance improves or is competitive in all cases.

| AUROC (Softmax) | | | | | | | | |
|---|---|---|---|---|---|---|---|---|
| **SVHN** | | | **CIFAR100** | | | **Tiny ImageNet** | | |
| **ResNet18** No L2 350 | No L2 60 | L2 60 | No L2 350 | No L2 60 | L2 60 | No L2 350 | No L2 60 | L2 60 |
| **Min** **0.893** | 0.791 | 0.854 | 0.846 | 0.841 | **0.862** | 0.842 | 0.857 | **0.869** |
| **Max** **0.938** | 0.879 | 0.931 | 0.858 | 0.869 | **0.880** | 0.870 | **0.887** | 0.885 |
| **Mean** **0.917**±0.013 | 0.838±0.029 | 0.888±0.026 | 0.852±0.004 | 0.855±0.008 | **0.872**±0.005 | 0.860±0.007 | 0.870±0.010 | **0.878**±0.005 |

| **ResNet50** No L2 350 | No L2 100 | L2 100 | No L2 350 | No L2 100 | L2 100 | No L2 350 | No L2 100 | L2 100 |
|---|---|---|---|---|---|---|---|---|
| **Min** **0.950** | 0.664 | 0.834 | **0.877** | 0.841 | 0.833 | **0.896** | 0.853 | 0.844 |
| **Max** **0.975** | 0.948 | 0.952 | 0.882 | 0.877 | **0.888** | 0.903 | **0.906** | 0.902 |
| **Mean** **0.964**±0.007 | 0.843±0.073 | 0.910±0.028 | **0.879**±0.002 | 0.860±0.010 | 0.859±0.016 | **0.899**±0.002 | 0.882±0.016 | 0.873±0.017 |

(b) Softmax performs worse in all cases versus GMMs on L2 normalized feature space with a singular exception: SVHN on ResNet 50.

Table 4: OoD detection results using (a) log probabilities from a GMM fitted over feature space and (b) softmax scores. ResNet18 and ResNet50 models were used, 15 seeds per experiment, trained on CIFAR10, with SVHN, CIFAR100 and Tiny ImageNet test sets used as OoD data. For all models, we indicate whether L2 normalization over feature space was used (L2/No L2) and how many training epochs occurred (60/100/350), and compare against baseline (No L2 350). There is no clear pattern of behaviour when using softmax scores for OoD detection, but using GMMs provides superior results.

## A.3   Fitting GMMs on Logit Space

A large amount of NC results in a feature space that contains most class information along $k = numclasses$ dimensions, a small subset of the total size of the feature space (e.g. $\mathbb{R}^{512}$ in ResNet18). This raises the question of whether a GMM could simply be fit over logit space, since it is in $\mathbb{R}^k$.

Table 4 shows the results of experiments with GMMs fit over logit space. This approach performs worse than GMMs fit over feature space in all cases. Intuitively, this makes sense: even under perfect NC, we

| | AUROC | | | | | | | | |
|---|---|---|---|---|---|---|---|---|---|
| | SVHN | | | CIFAR100 | | | Tiny ImageNet | | |
| **ResNet18** | No L2 350 | No L2 60 | L2 60 | No L2 350 | No L2 60 | L2 60 | No L2 350 | No L2 60 | L2 60 |
| Min | **0.867** | 0.831 | 0.846 | 0.782 | 0.767 | **0.848** | 0.769 | 0.759 | **0.864** |
| Max | **0.932** | 0.883 | 0.928 | 0.807 | 0.801 | **0.871** | 0.815 | 0.805 | **0.890** |
| Mean | **0.897**±0.017 | 0.846±0.013 | 0.885±0.021 | 0.797±0.007 | 0.785±0.011 | **0.860**±0.006 | 0.798±0.014 | 0.780±0.014 | **0.879**±0.008 |
| | | | | | | | | | |
| **ResNet50** | No L2 350 | No L2 100 | L2 100 | No L2 350 | No L2 100 | L2 100 | No L2 350 | No L2 100 | L2 100 |
| Min | **0.908** | 0.682 | 0.838 | **0.827** | 0.764 | 0.826 | 0.843 | 0.770 | **0.846** |
| Max | **0.947** | 0.909 | 0.927 | 0.842 | 0.817 | **0.872** | 0.874 | 0.837 | **0.883** |
| Mean | **0.929**±0.011 | 0.834±0.054 | 0.899±0.023 | 0.834±0.004 | 0.795±0.014 | **0.850**±0.013 | 0.855±0.007 | 0.811±0.020 | **0.868**±0.012 |

Table 5: OoD detection results for ResNet18 and ResNet50 models using log probabilities taken from GMMs fitted over logit space instead of feature space (same experimental setup as Table 4). This approach performs worse in all cases versus using GMMs on L2 normalized feature space (see Table 1).

would expect OoD inputs to increase the variability of class clusters in arbitrary dimensions of feature space. A Singular Value Decomposition (SVD) over feature space supports our intuitions. In Figure 6, we show the SVD of all training embeddings for CIFAR10, along with the singular values for the test set and SVHN OoD test set projected onto the the same basis used for the training singular values. As we would expect, the first 10 singular values contain nearly all information. However, the latter 502 singular values contain significantly more information in the OoD case. This information is critical to identifying OoD examples in feature space and, due to dimensionality reduction, is severely reduced in logit space.

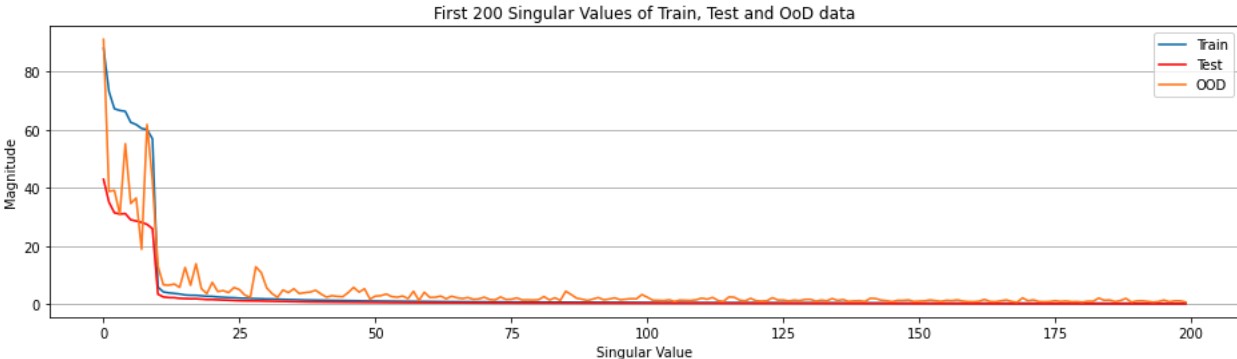

Figure 6: The first 200 singular values for CIFAR10 train and test sets, as well as the SVHN OoD set. As expected, singular value magnitudes fall off drastically after the first 10. Singular value magnitudes for OoD examples remain higher after 10, indicating greater deviation from the Simplex ETF structure. This information is exploited by the GMM to identify OoD examples, and is much less prevalent in the heavily reduced dimensionality of logit space.

## A.4 Overtraining with L2 Normalization

Table 6 shows the results of overtraining with L2 Normalization (L2 350). While there is not a substantial penalty for overtraining by 10 to 100 epochs (Figure 5, Right), training for the full 350 epochs (as with the DDU baseline) starts to reduce OoD performance by a few percentage points. We note that there is a tradeoff with accuracy, which does increase when overtraining to 350 epochs.

| | AUROC | | | | | | | | | | | |
| --- | --- | --- | --- | --- | --- | --- | --- | --- | --- | --- | --- | --- |
| | SVHN | | | | CIFAR100 | | | | Tiny ImageNet | | | |
| **ResNet18** | No L2 350 | L2 350 | No L2 60 | L2 60 | No L2 350 | L2 350 | No L2 60 | L2 60 | No L2 350 | L2 350 | No L2 60 | L2 60 |
| **Min** | 0.877 | 0.879 | 0.848 | **0.924** | 0.861 | 0.870 | 0.796 | **0.878** | 0.881 | 0.885 | 0.809 | **0.898** |
| **Max** | 0.949 | 0.921 | 0.953 | **0.961** | 0.885 | 0.879 | 0.845 | **0.886** | 0.909 | 0.902 | 0.872 | **0.911** |
| **Mean** | 0.915±0.018 | 0.904±0.011 | 0.912±0.035 | **0.938**±0.010 | 0.875±0.008 | 0.874±0.002 | 0.824±0.014 | **0.881**±0.002 | 0.894±0.009 | 0.892±0.004 | 0.841±0.015 | **0.904**±0.004 |
| | | | | | | | | | | | | |
| **ResNet50** | No L2 350 | L2 350 | No L2 100 | L2 100 | No L2 350 | L2 350 | No L2 100 | L2 100 | No L2 350 | L2 350 | No L2 100 | L2 100 |
| **Min** | 0.903 | 0.903 | 0.869 | **0.927** | 0.817 | 0.876 | 0.794 | **0.892** | 0.881 | 0.880 | 0.852 | **0.912** |
| **Max** | **0.987** | 0.943 | 0.98 | 0.955 | 0.891 | 0.891 | 0.866 | **0.896** | 0.909 | 0.898 | 0.905 | **0.918** |
| **Mean** | **0.955**±0.026 | 0.926±0.013 | 0.944±0.029 | 0.945±0.007 | 0.871±0.018 | 0.886±0.005 | 0.837±0.022 | **0.894**±0.001 | 0.894±0.020 | 0.888±0.005 | 0.880±0.016 | **0.915**±0.002 |

(a) OoD performance begins to decay substantially when models are overtrained with L2 normalization, in line with our discussion in Section 4.2.2.

| | Accuracy | | | | | | | | |
| --- | --- | --- | --- | --- | --- | --- | --- | --- | --- |
| **ResNet18** | No L2 350 | L2 350 | No L2 60 | L2 60 | | **ResNet50** | No L2 350 | L2 350 | No L2 100 | L2 100 |
| **Min** | 0.928 | **0.936** | 0.881 | 0.924 | | **Min** | 0.930 | **0.947** | 0.896 | 0.937 |
| **Max** | **0.941** | **0.941** | 0.912 | 0.929 | | **Max** | 0.946 | **0.952** | 0.927 | 0.943 |
| **Mean** | 0.935±0.005 | **0.938**±0.002 | 0.898±0.008 | 0.927±0.001 | | **Mean** | 0.939±0.006 | **0.949**±0.002 | 0.916±0.008 | 0.941±0.002 |

(b) Accuracy increases slightly when substantially overtraining with L2 normalization, but OoD performance drops. For ResNet50, higher accuracy is achieved in only 100 epochs compared with the baseline.

Table 6: OoD detection (a) and classification accuracy results (b) for ResNet18 and ResNet50 models, 15 seeds per experiment, trained on CIFAR10, with SVHN, CIFAR100 and Tiny ImageNet test sets used as OoD data. For all models, we indicate whether L2 normalization over feature space was used (L2/No L2) and how many training epochs occurred (60/100/350), and compare against baseline (No L2 350).

## A.5 Neural Collapse Measurements for NC Loss Intervention

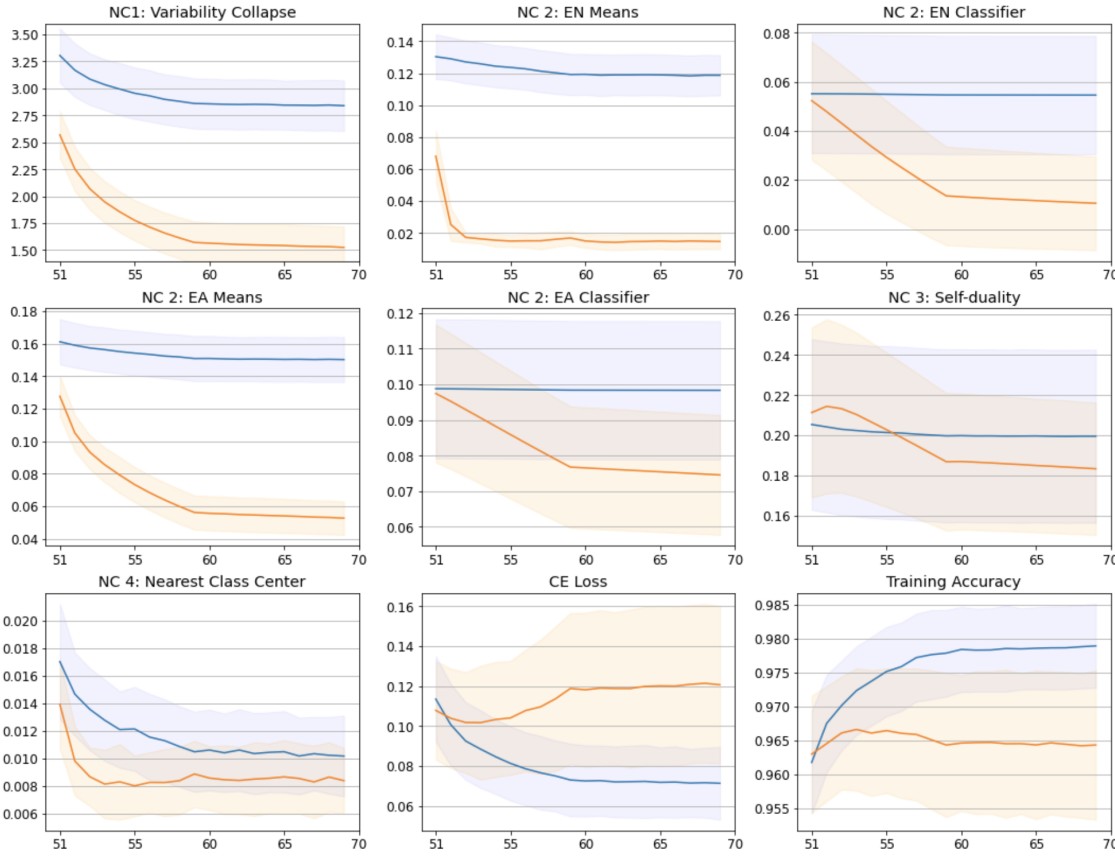

Figure 7: Mean (solid lines) ResNet18 NC measurements from the NC loss (see Equation 3) experiment over 15 model seeds, along with CE loss and classification accuracy. Shading around the solid line shows standard deviation. Blue is the control group, orange is the intervention group. The intervention started at epoch 50, and proceeded for 20 epochs. Over the same training period, the intervention group has substantially more NC, while CE loss and training set classification accuracy are relatively unchanged. This indicates that NC and CE loss effects were controlled successfully.

## A.6 Additional Figures

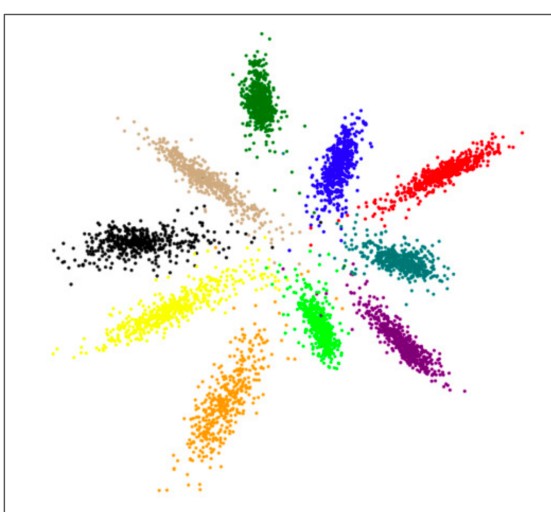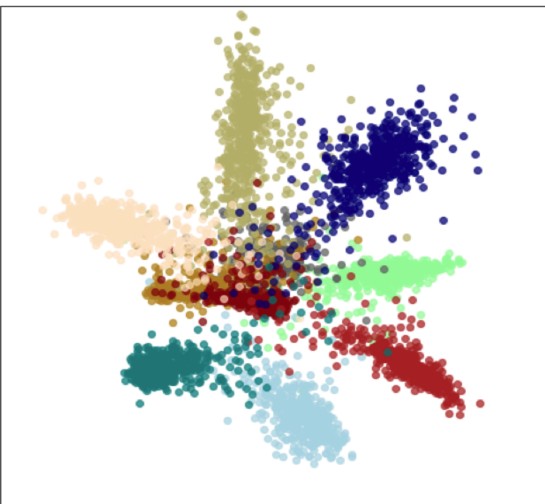

Figure 8: Building intuition about embedding spaces: Training models for low cross-entropy or high accuracy does not necessarily structure embedding space in an optimal way for OoD detection. Left: MNIST training features in a 2-dimensional feature space. The only structure explicitly required by cross-entropy loss is that features are linearly separable. Right: CIFAR10 training features in a 2-dimensional feature space. The network finds it more difficult to make these more complex features linearly separable, but even with spectral normalization, the structure is still arbitrary. In both cases, the within-class variance could be reduced substantially, preserving appropriate notions of sensitivity and smoothness while not affecting classification. These tighter class means could be separated to allow OoD features more inter-class distance to fall away from fitted GMM modes.

