# OpenReview forum: "Linking Neural Collapse and L2 Normalization with Improved Out-of-Distribution Detection in Deep Neural Networks"
_TMLR — Accepted by TMLR_

### Review · Reviewer_11H5 · 2022-10-16

**Summary Of Contributions:**

The authors look at the problem of OOD detection and apply L2 regularization over the feature space and demonstrate some improvements in speedup and AUROC when combining with DDU.  The authors also investigate the relationship between neural collapse and OOD detection performance.

**Audience:**

Yes

**Claims And Evidence:**

Yes

**Requested Changes:**

Critical adjustments:
- I really think that major revisions should be made to the writing of the paper.  Outside of reading the introduction, I really had no idea what the authors contributions are throughout reading the paper.  There's also a lot of extraneous details about experimental setup and additional background info that is neither used as motivation nor used to explain the methodology.  The paper severely lacks a section that explains what the authors introduce (L2 regularization on the feature space) and motivate why the authors think this would improve performance.  Similarly there isn't anything motivating the study of neural collapse and connections with OOD performance, the authors just define neural collapse and move on to explain experimental setup.
- More baselines for OOD, more datasets for NC experiments, can also look at metrics like AUPR
- Some statements such as "evidence of a tightening lower bound" should be explained
- the final conclusion of the study of NC and OOD performance ends with the conclusion that more NC sometimes helps OOD performance and that sometimes it doesn't but it isn't clear when this occurs.  The authors leave this as an open question, but given that part of the contributions is a study of the relationship between NC and OOD performance, I think there should be more work done to investigate this.

Minor comments:
- Could the authors format some of the citations to use parenthetical citations (\citep{}) instead of in-text citations (\citet{}) (specifically where I should not be reading the authors name as part of the sentence)?  That would greatly improve the legibility of the paper
- Would also help to give timing results on how long training takes for the L2 regularized model vs DDU without L2 regularization

**Strengths And Weaknesses:**

Strengths:
- Interesting problem

Weaknesses:
- Problem definition in section 2.1 is not very clear.  What does it mean in the sentence before Eq 1 "For OOD detection, we can conceive of the model as learning a data generating distribution $p(y|x)$..."?  Should it be $p(x, y)$? It also isn't clear to me how this formulation is useful for OOD detection, and Eq 1 is not referenced later in the paper.  It also isn't clear to me how this is related to the sentence that follows: "Under this regime, we desire that the model learns a density over feature space..."  I think it would also help to clarify the definition of OOD too, right now the text just says that $X_{in}$ contain inputs drawn from the same distribution as the training data and $X_{out}$.  I think it should be clear that the distributions that $X_{out}$ and $X_{in}$ are sampled from have different supports.
- Missing some related works for OOD detection: ie)
 Hendrycks, Dan, and Kevin Gimpel. "A Baseline for Detecting Misclassified and Out-of-Distribution Examples in Neural Networks." (2016).
Liang, Shiyu, Yixuan Li, and R. Srikant. "Enhancing The Reliability of Out-of-distribution Image Detection in Neural Networks." International Conference on Learning Representations. 2018.
- Paper organization can be improved.  Section 2 is a very long background section and immediately jumps into Section 3 which starts off with experimental setup and mainly describes **how** the authors do certain things **but not why**.  It would be more helpful to shorten the background section and use it to motivate what the authors are trying to do and describe how the experiments answer those questions, and then transition into Section 3.  I also think that a large portion (specifically details on how models are trained) of Section 3.1 can be moved to the Appendix.  The main text should focus on describing which architectures, datasets, and baselines you use so that the readers can understand the experimental results and figures.
- Not sure what is meant by "evidence of a tightening lower bound" in section 4.2, what lower bound?
- Legend in figure 4 isn't particularly useful.  Overall figures 3 and 4 have a lot going on because 15 seeds are used, I'm not exactly sure what I should be looking at.  Maybe the authors can just plot the mean and variances?
- Should include results for additional baselines for OOD: authors only compare to no L2 regularization

---

### Review · Reviewer_jLid · 2022-10-21

**Summary Of Contributions:**

The paper presents an observation that L2 normalization of features improves OoD detection performance and/or enables the networks to reach better OoD performance in a short time. Additionally, they empirically show a correlation between neural collapse (NC) and OoD performance. Experiments are performed on cifar, svhn and tinyimagenet datasets with resnet architectures.

**Audience:**

Yes

**Claims And Evidence:**

No

**Requested Changes:**

Please address the writing comments and provide intuitions and/or justifications for 1) why L2 normalization might improve NC and OoD performance and 2) when better NC would lead to better OoD performance.

**Strengths And Weaknesses:**

## Strengths

1. The observation that L2 normalization has a positive effect on OoD performance is interesting and would be easily adopted.

2. Correlation between NC and OoD performance is interesting but warrants further investigation.

## Weaknesses

1. No clear justification or explanations. For instance, there is no clear intuition/explanation provided for why L2 normalization yields NC. Also, it is not clear why NC would lead to better OoD improvements. I think the empirical observations would be strengthened if proper intuitions and/or explanations are provided. Specifically, NC --> OoD connection is problematic without any justifications as it is observed in the last para before sec. 5 that "more NC is not always better" in contrast to the rest of the paper.

2. Writing lacks clarity. In many places, the writing is not clear and lacks clear explanations.
a) In Sec 2.1, the function definition should be a mapping from a set to another set, X \in X_in doesn't make sense (should it be X\subset X_in?), and Eq.1 doesn't make sense (should it be P(y|x \in X_in \cup X_out)?). Also please introduce all the symbols in all equations.
b) In Sec. 2.2, what is a bi-lipschitz constraint? Please cite.
c) NC1 would be zero if the cross-entropy is zero right? Please clarify this.
d) what is the form mutual coherence matrix?
e) For example, what is meant by "global means of classes" (u_G)?
f) legend for fig. 3

---

### Review · Reviewer_WGXv · 2022-10-22

**Summary Of Contributions:**

This paper tackles an important problem of out-of-distribution detection with neural networks. The authors of the paper build upon the previously proposed method of "deep deterministic uncertainty", and demonstrate that the use of $L_2$ regularization over the feature space can serve as a simple yet effective method to improve the method, both in terms of the out-of-distribution detection performance and in terms of the training time needed. The authors of the paper demonstrate that such improvement can be attributed to the fact that the L2 regularization induces early neural collapse of the model. Such neural collapse is experimentally demonstrated to be helpful for OoD detection.

**Audience:**

Yes

**Broader Impact Concerns:**

I don't see any concerns regarding this.

**Claims And Evidence:**

No

**Requested Changes:**

First of all, it would be great if the authors of the paper can address my concerns listed above about the method. In addition, I also find the structure of the method section to be a little bit confusing. It might be better to first mention the proposed L2 regularization first before talking about experimentation etc..

**Strengths And Weaknesses:**

Strengths:
- The authors of the paper conduct careful experiments to demonstrate the effect of L2 regularization on the neural collapse phenomenon.
- The proposed method is simple yet seems to be effective.
- The proposed method seems to be well-motivated.

Weaknesses:
- I find the description of the proposed L2 regularization to be very confusing. What is exactly the formula of the L2 regularization? Is Equation 9 of the paper the regularizer used? What is the hyper-parameter associated with this additional term? Also, why would L2 regularization over feature space "constrain the feature vector to a point on the surface of a hypersphere"? I feel like there is a lot of important but used explanation missing regarding the proposed method.
- I still don't see a causal connection between the neural collapse and improved OoD performance. Why is NC helpful for the task of OoD detection? While the authors of the paper provide experimental evidence for this, it would be great if there is some additional explanation/derivation from a theoretical standpoint.

---

### Decision · Action_Editors · 2022-12-02

**Recommendation:** Accept with minor revision

**Comment:**

The paper tackles an important topic (OoD), proposes a simple and promising looking method (L2 normalization), and establishes an interesting link with Neural Collapse (NC). We thank the authors for their engagement in the rebuttal process, and note that the paper was substantially improved during the process.

However, I think the current version requires some minor revisions before acceptance. In particular, while a link with NC is established, this relationship between OoD and NC is unclear, and whether this plays a major role in L2's improved OoD performance is unclear. I would like to request the following minor revisions before acceptance:

1) The title currently "Inducing Early Neural Collapse in Deep Neural Networks for Improved Out-of-Distribution Detection". Please soften the implication in the title that inducing early collapse is the main cause of improved OoD. The evidence shows the L2 improves OoD, and OoD and NC are related, but there is not sufficient support for this title.

2) Soften contribution (4). The paper can claim to uncover an interesting link between early NC and improved OoD detection. But I think the claim "NC encourages OoD detection under our proposed modification to the DDU method" is slightly too strong.

3) Please add an experiment that shows the performance of L2 with a large number of epochs (the same number as used in the baseline in Table 1). Even if the results are not good, this would help practitioners understand that this method is most beneficial in conjunction with short schedules.

The following is optional, and not required for acceptance, but I think would substantially strengthen the paper:

4) Add an experiment using L2, training on one dataset other than CIFAR10, ideally a larger dataset (e.g. ImageNet), where training cost is more likely to be a bottleneck, and training with a short schedule would be most beneficial in practice.

A couple of trivial changes needed:

* After the refactor, some concepts (e.g. NC2 on P5) are referenced before definition.
* There are still instances of "L2 regularization" instead of "L2 normalization" which caused confusion in the reviews.
* I saw one typo (standad->standard P2)

**Audience:**

The paper addresses an important topic, OoD detection, that is recognized by the reviewers. It also proposed a very simple architectural change (L2 norm the output features), that could improve performance. This paper might be of interest to practitioners wanting to improve OoD performance. However, it would be useful for practitioners to see experiments with L2 and a longer training duration. Evidence from training on one other dataset other than CIFAR10 could also increase interest, ideally one where training speed is more likely to be an important bottleneck.

For researchers, the link with NC could open interesting further study along these lines.

**Claims And Evidence:**

The paper makes four claims:

1) L2 normalization in features space matches or improves OoD detection & classification performance compared to the [DDU](https://arxiv.org/abs/2102.11582) algorithm.

2) L2 normalization yields performance benefits in reduced training time compared to DDU.

3) L2 normalization yield faster [Neural Collapse (NC)](https://arxiv.org/abs/2008.08186) than without it.

4) NC encourages OoD detection under their modification, and plays a role in achieving better OoD with less training. Plus the connection to ETFs (a consequence of NC) permits an elegant analytical framework for further study of mechanisms that govern uncertainty and robustness.

The reviewers generally agree that (1-3) are reasonably well substantiated. The results on the DDU benchmark are generally positive. There are, however, instances where their method (L2) underperforms the baseline with longer training both in terms of accuracy and OoD in the average case. The paper does not present L2 with a longer training schedule, it would be useful to know whether L2 continues to match/outperform DDU in this setting. The experiments are limited to training on a single dataset (CIFAR10), which limits the strength of the evidence.

All reviewers raised issues with (4). After the rebuttal, the writing was substantially improved, and a more detailed motivation was provided. However, the link between L2 and improved OoD being a consequence of NC seems unclear. It seems well supported that L2 induces faster NC than baseline (Tab 3). However, with longer training NC and OoD appear anti-correlated. Further, directly optimizing NC can decrease OOD after an initial period. One of the concluding sentences recognizes that this is still an open question "this harmonizes with our intuition that NC is only one of several factors acting to condition feature space for OoD detection. These factors are all susceptible to the variability of training dynamics, as well as situational dependencies such as architectures and datasets.". Claim (4) and the title are still slightly too strong.